# Remotely Controlled Electronic Goalkeeper: An Example of Improving Social Integration of Persons with and without Disabilities

Massimiliano Donati , Federico Pacini , Luca Baldanzi , Mauro Turturici and Luca Fanucci *

Department of Information Engineering, University of Pisa, 56122 Pisa, Italy;
massimiliano.donati@unipi.it (M.D.); federico.pacini@phd.unipi.it (F.P.)
* Correspondence: luca.fanucci@unipi.it

**Abstract:** Social integration is an essential part of the life of every human being, but for people with disabilities, there are many situations in which it is still very low. For instance, in sports and outdoor gaming, there is a barrier between players with and without disabilities. Individuals with disabilities play sports almost exclusively with disabled players, not only during official events such as Paralympic games but also in occasional sports groups, while the mixing of people with and without disabilities in sports activities is a key factor of social inclusion. In order to allow a person with motor-skill impairments to play on the same ground as their non-disabled peers, we developed a novel piece of Assistive Technology that lets a person with motor-skill impairments to control a system acting as a goalkeeper during a non-professional football match, with approximately the same performances as a goalkeeper without motor-skill impairments. This electro-mechanical system is composed of a three-meter-long metal guideline and a human-shaped dummy sliding along it. The system is equipped with a high-torque battery-powered direct-current motor and it is controlled by means of electronic boards and sensors to ensure safety and good usability also for players with severe mobility impairments. The results of a pilot testing demonstrated the robustness and high degree of usability of the system, enabling people with motor-skill impairments to competitively participate in matches with non-disabled peers.

**Keywords:** assistive technologies; disability; motor impairments; football; goalkeeper; game; social integration; inclusion; sport; outdoor

## 1. Introduction

Social inclusion of individuals with disabilities is defined as the possibility of having full and fair access to activities, social roles, places, and relationships alongside non-disabled people [1]. In recent decades, such inclusion has undergone significant improvement [2,3], largely due to technological and societal advancements that have enabled people with physical, sensory, or intellectual disabilities to live more independently and participate more fully in work and social activities. However, it is worth noting that social inclusion is an ongoing process, and there is always room for improvement. Despite the progress made, achieving full social inclusion in all environments, including schools, workplaces, entertainment, and others, requires continued efforts to address barriers that prevent individuals with disabilities from fully participating in society.

Many organizations claim more attention to people with disabilities and ask governments to take specific actions towards social inclusion, for instance to remove architectural barriers [4], advocate for disability rights, encourage socialization and community engagement, etc. As a result, in many developed countries, people with disabilities reached a higher social status and specific laws to protect their rights. One of the best outcomes is the UN Convention on the Rights of People with Disabilities (CRPD) of 2007 [5], which gives important guidelines to promote the full realization of all human rights and fundamental

freedoms for all persons with disabilities, also through the development and use of new technologies (art.4).

Technological solutions and digital technologies entail major opportunities to promote and facilitate the social integration of people who live with different types of disabilities [6,7]. Assistive Technology (AT) plays a fundamental role, as it can in many cases fill the gap between the needs and capabilities of people with disabilities. Thanks to AT, also persons with severe impairments can move, get around, communicate, use a computer, and perform tasks that might otherwise be difficult or impossible; however, we believe that research and development of AT must be broadened to more human contexts.

In particular, there is a general lack of social integration in outdoor gaming between players with and without disabilities, particularly among youths, for whom socialization is in general more important than for adults [8]. Participation in sports is a good chance to improve social engagement that can improve well-being [9,10]. Sometimes, this lack of participation is due to the poor awareness of those without disabilities on how to include peers with disabilities in sports [11], fear, or parental behavior [12], but in most cases, it is due to different physical capabilities [12]. For instance, a person with motor-skill disabilities can play sports such as football or basketball with more difficulties with respect to their non-disabled peers and playing together on the same ground may result in a game too hard and demoralising for the player with disabilities and not too competitive for the one without disabilities. Despite attempts to assess the needs of people with disabilities for physical activities and sports [13,14], and propose variant of traditional sports such as basketball with all players sitting on a wheelchair to include young people with disabilities [15], at present, the practice of sports for people with disabilities remains uncommon and mainly takes place in dedicated events.

This paper presents a novel system that allows a person with motor-skill disabilities to be competitive in an outdoor gaming situation, on the same ground as their peers without disabilities, playing the role of the goalkeeper in a non-professional five-a-side football match, which instead requires high-speed reaction and coordination of all limbs. We addressed five-a-side football, because this type of game is the most popular in non-competitive matches and because it guarantees a smaller size for the entire system, facilitating its transportation. Moreover, we chose the role of the goalkeeper because it can be emulated with a reduced number of degrees of freedom so that it can also be played theoretically with good performances by a child with severe disabilities.

To the best of our knowledge, the proposed system is completely new in the AT landscape. Other non-human gatekeepers we found in the literature are completely automated systems with no or reduced possibility of user control [16,17]. They are mainly designed to save each goal autonomously and are usually used for recreational purposes or training of professional players. Furthermore, solutions present on the market have the same characteristics and objectives (e.g., Goalias [18] by IAS and Robokeeper [19]). All these devices feature a set of cameras looking at the ball and advanced algorithms to compute [20,21] and actuate [22] the best dummy positioning with respect to the incoming ball. Thus, none of these systems can be employed for the purpose of enabling a human to interact directly with the game in the role of goalkeeper.

## 2. Materials and Methods

The idea behind the proposed Electronic Goalkeeper (ElGo) device is to enable a person with motor-skill disabilities to play the role of the goalkeeper in a five-a-side football match. Since the goalkeeper's most important task is to parry the kicks toward the goal, the system is conceived as a physical barrier, made by a human-shaped dummy, which can be remotely controlled to move along the goal line by means of a simple user interface that allows the exploitation of the player's residual motor skills (see Figure 1). In addition to usability, safety conditions during use and performance in terms of responsiveness and robustness of the system are equally important aspects to consider for ensuring competitiveness on the football ground. This leads to an exciting and challenging user experience both for the

goalkeeper user with disabilities and for all the players without disabilities involved in the match.

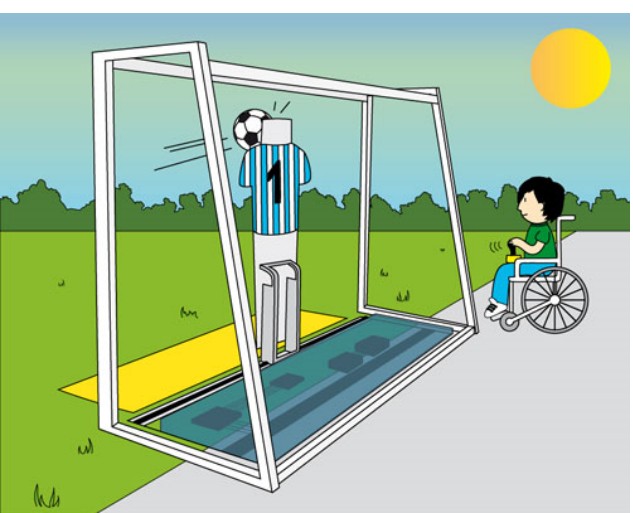

**Figure 1.** ElGo system concept.

Our design approach was completely user-centered [23], with particular reference to its declination in assistive technologies development [24,25]. End users were involved throughout the entire design and development process, to create an accessible and highly usable system for them. In particular, we mainly focused on the users' motor difficulties and their needs, but also on the application context and the task of the ElGo system.

*2.1. System Requirements*

The starting point was a deep understanding of the users' needs and the expected use cases of the system. Furthermore, aspects given by the football game itself and by the interaction of users with the goalkeeper were taken into consideration. This initial phase included several interviews with potential end users and helped to define the general requirements of the system and the desired features.

The analysis conducted showed that the ElGo system must be quick enough in terms of responsiveness to the user commands and speed of the dummy to ensure the same probability as a non-professional goalkeeper of saving a goal, and ultimately to be competitive on the football ground. In addition, the entire mechanical infrastructure must be hard enough to face strong kicks done by a skilled player without incurring deformations or damages (i.e., robustness).

One of the key aspects arising from the analysis is usability. From this point of view, the ElGo system must be very intuitive and easy to use, allowing it to be operated by means of a large variety of switch sensors adapted for people with motor-skill disabilities and an extremely reduced set of inputs, even only two. At the same time, the system must provide a flexible and adaptable user experience with different gaming modes characterized by an increasing number of degrees of freedom in controlling the dummy, allowing the person to exploit their functional capabilities.

Furthermore, the system must be safe for all players during the game (i.e., safety), both from mechanical and electrical points of view. In particular, because of the moving parts that could bump players, for example when approaching the dummy or while picking up the ball from the goal after scoring, there must be a procedure to automatically stop the movement immediately when a person is detected in the proximity of the system. In addition, mechanical and electronic components and interconnecting cables must be protected with appropriate cases.

Finally, the ElGo system should be easily transportable from one playing field to another, and its setup should not require special technical skills. These requirements call

for a dismountable mechanical structure composed of lightweight elements that can be quickly assembled together and battery-powered electronic components.

### 2.2. Preliminary Analysis

As previously stated, the most important feature of the ElGo is to allow a user with motor-skill disabilities to match against other players at the same level, and thus, the responsiveness of the system and the velocity of horizontal sliding of the dummy must be comparable to the performance of a non-professional goalkeeper without disabilities. In particular, ball speed and weight are two of the principal factors to consider for the definition of the system performance, as they are closely related to the sliding velocity expected for the dummy. They are also fundamental in designing the damping and impact resistance capabilities of the mechanical structure.

The following subsections summarize the most important preliminary analysis carried out during the design phase in order to investigate the required system performance and to identify the main components suitable for achieving it.

### 2.2.1. Kick Speed

We carried out an initial analysis of kick speed by non-professional players, focusing in particular on boys from 14 to 20 years old; approximately the same age range of target end users. We focused on males, since they achieve on average greater ball speed than females [26]. The test involved five volunteers with different skills, ages, and weights. The boy classified as "skilled" was a person playing in an agonistic soccer team, the "medium skilled" one was playing soccer frequently for recreational purposes, and the others were not playing soccer at all or rarely. They were each asked to perform 30 penalty kicks from different positions and distances from the goal. We recorded the scene with a good commercial camera at a resolution of $1280 \times 720$ at 120 fps and then calculated the time of flight of the ball for each attempt. Table 1 shows the characteristics of the players and the average kick speed achieved by each of them during the test session.

**Table 1.** Average kick speed reported by young non-professional football players.

| Player/Skill Level | Age (years)/Weight (Kg) | Average Kick Speed at Penalty (m/s) |
|---|---|---|
| 1/skilled | 17/62 | 11,5 |
| 2/medium skilled | 18/67 | 9,8 |
| 3/non-skilled | 16/58 | 7,1 |
| 4/non-skilled | 16/56 | 8,2 |
| 5/non-skilled | 14/61 | 6,8 |

In addition to our experimental data, we found that a professional footballer can kick a soccer ball at a speed of about 30 m/s, while an adult amateur can kick one at a speed of about 22 m/s [27]. Ultimately, we decided to take a value of 20 m/s as the maximum incoming velocity of the ball toward the goal, which is a safe assumption, since about twice the value measured in the test with the target users and near the average performance of non-professional adult footballers. Such a defined value was later used as an input parameter to set the maximum dummy displacement speed and to size the damping system of the metal structure.

### 2.2.2. Processing Delay

Human reaction time is an important factor to compute the total amount of time to perform a body movement [28]: in our case, to control ElGo and try to save the incoming ball toward the goal.

Since a person with a motor-skill disability is often affected also by neurologic-related issues, we considered that our end users can have a longer reaction time than other players. For this reason, the entire processing chain must be as fast as possible and reactive to user

inputs. To this end, we chose a real-time embedded system, based on a microcontroller and a simple event scheduler to manage the entire signals elaboration chain. This low-complexity firmware architecture also ensures high reliability, low power consumption, and quick development time compared to platforms featuring Real-Time Operating Systems (RTOS).

In particular, the processing time must be computed as a part of the system reaction time, and thus, must be added to the human reaction time [29]. In other words, ElGo must be faster than a non-professional goalkeeper in order to account for the processing delay and the impact of the user reaction time. Since the latter time is several orders of magnitude higher than system processing and communication times, there is no need to take into account time varying delay techniques [30] in this context.

### 2.2.3. Dummy Displacement and Simulation

Based on the foregoing considerations, we estimated that approximately 1 s must be the time for the dummy to move from one side to the other of the goal (i.e., system reaction time). Taking into consideration the dimensions of a five-a-side football regular goal (3 m long, 2 m tall, and 1 m deep), this resulted in an average required velocity of 3 m/s for the dummy. We did not state any conditions about its acceleration.

Considering the aforementioned constraint and the expected weight of the mechanical infrastructure, we run several simulations in order to compute the motor size, particularly the instant power and the torque. These simulations were carried out using Matlab Simulink and using some parameters taken from different off-the-shelf DC motors as input. Considering a 24 V DC motor capable of delivering a continuous power of 700 W and a mechanical torque of 4.5 Nm, the 3-meter displacement of the dummy occurs in less than 1 s, with a maximum velocity and maximum acceleration of 6.2 m/s and 13 m/s$^2$, respectively, [31].

In addition, the same simulation software tool was used to identify optimal characteristics of the damping system needed to cushion the impacts of the ball in order to guarantee minimum rebounds and prevent the onset of vibrations that could damage the whole system. Specifically, such a damping system was designed to absorb the impact of a ball with a velocity of 20 m/s, limiting the amplitude of the produced oscillations to less than 0.05 m [31].

### 2.2.4. Design of User Experience

As previously mentioned, the ElGo system must be easy to use, providing a satisfying user experience that is tailored to the player's individual capabilities. To achieve this objective, we designed three game modes for the initial release, each offering varying levels of complexity and degrees of freedom. This allows any user to choose their favorite and most suitable game mode.

We set a basic mode, with a single degree of freedom (i.e., left and right) and fixed speed, suitable for players with severe motor-skill impairments. This mode can be operated with two switches only, for instance two big buttons. Assuming, without loss of generality, physical buttons are placed on a plane, the idea is that by pressing the left button, the dummy should move leftwards as long as the user continues pressing it; vice versa for the right button. The second game mode uses four switches and two speed grades for both directions. Considering again a plane with four buttons displaced as two per side, one above the other, the left side of the plane is responsible for moving the dummy leftwards, and this time, the user can decide to go for slower or faster movement. The last mode features 16 speed grades for both directions, which are selected by a resistive analog joystick, whose value is acquired by an Analog-to-Digital Converter (ADC). The way it works is as follows: by dividing the joystick's movement area into two sides, i.e., Cartesian quadrants I and IV, representing the right side together, and quadrants II and III, representing the left side, the user decides the direction of the dummy by moving the stick to the right or left side; the intensity is then calculated by projecting the vector indicated by the stick onto the X axis of the plane and quantifying its value in 16 buckets (see Figure 2).

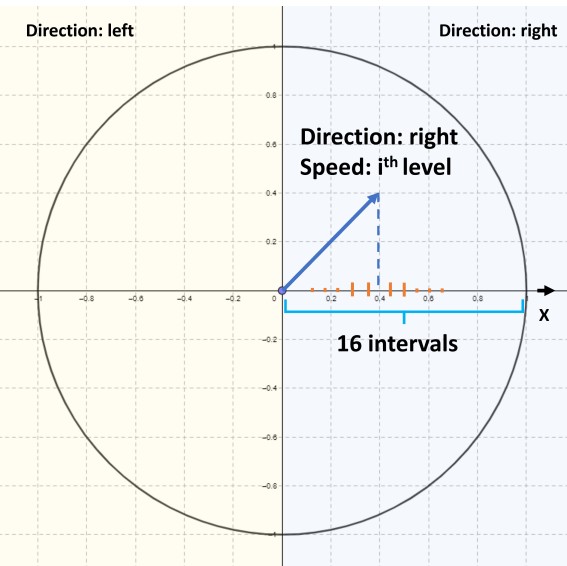

**Figure 2.** Joystick movement space representation for the third mode of functioning.

Since the objective was to design a human–machine interface as inclusive as possible, the idea behind the first two games mode is that it is not mandatory to use buttons to operate the dummy. The concept is to use a way to express a Boolean value, and thus, the controller device can be whatever interface commercially available that is able to electrically behave as a switch and be handled by the user's residual capabilities. Examples of possible interfaces are push buttons with low activation forces or large sizes, sip-puff sensors, finger buttons, and many more [32]. A possible enhancement could be to tune the speed of the dummy depending on the user's responsiveness. Nowadays, Neural Network models and Reinforcement Learning (RL) approaches are popularly used for fine-tuning system control. RL is useful because it allows to take into consideration also external disturbance [33,34].

*2.3. System Architecture*

The overall architecture of the proposed ElGo system is basically composed of a mechanical infrastructure to support the sliding dummy, an electromechanical motion transmission subsystem, and an electronic subsystem responsible for controlling the entire system and ensuring safe conditions during use.

The following subsections provide a detailed description of these components.

2.3.1. Mechanical Infrastructure and Transmission Subsystem

The heart of the mechanical infrastructure is represented by a commercial aluminum guideline and a plate of the same material sliding on it. The chosen architecture ensures high reliability for such a core component of the system and also ensures an easy replacement in case of failure. In addition, a custom-designed aluminum framework is bolted to the plate over the guideline. It includes a damping system consisting of two opposed springs and supports for attaching the dummy. The latter is made of transparent polycarbonate, thus ensuring both robustness and lightness and also allowing players to see incoming kicks while remaining behind the goal. The choice of making the metal parts from aluminum is due to its excellent mechanical strength vs. the reduced overall weight.

The transmission subsystem is made of the DC motor powered by two automotive lead-acid 12 V batteries, a gear wheel, a tension pulley, and a plastic belt reinforced with steel filaments. The belt must be properly tense; otherwise, the high torque of the DC motor could cause it to slide along the pulley, and perhaps permanent damage to the mechanical components. To this aim, an adjustable tension sheave is installed; this can be released with a dedicated secure handle, if necessary. This subsystem also includes two inductive sensors

mounted near its edges and an optical encoder used by the electronic subsystem to detect the end of stroke and to track the position of the dummy, respectively.

A computer-made scale model of the entire mechanical infrastructure, mounted in a door (3 m large and 2 m tall), is shown in Figure 3a. The semi-transparent plastic case for protecting any mechanical and electrical part from incoming kicks is also shown. The surface area of the dummy is comparable to that of a human body with slightly open arms. Figure 3b shows a detailed model of the transmission subsystem.

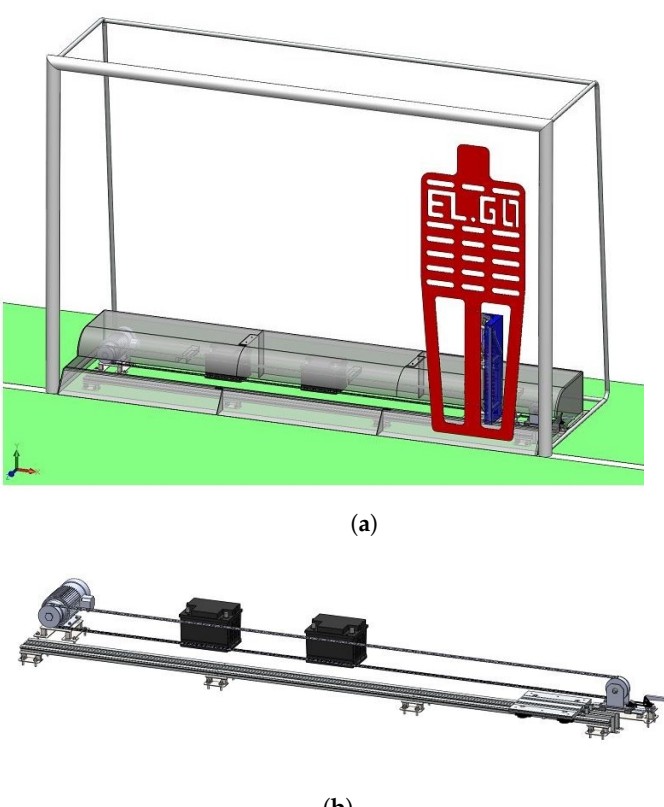

(**a**)

(**b**)

**Figure 3.** Computer-made models of the ElGo system: (**a**) mechanical infrastructure and (**b**) transmission subsystem.

### 2.3.2. Electronic Subsystem

The electronic subsystem is composed of four separate boards in charge of specific control tasks implemented by dedicated firmware. Its functional block diagram is shown in Figure 4. The circuits are powered by means of a 5 V DC power supply or 4.8 V battery pack (four AA-sized rechargeable batteries) to also ensure operation without the availability of a power plug.

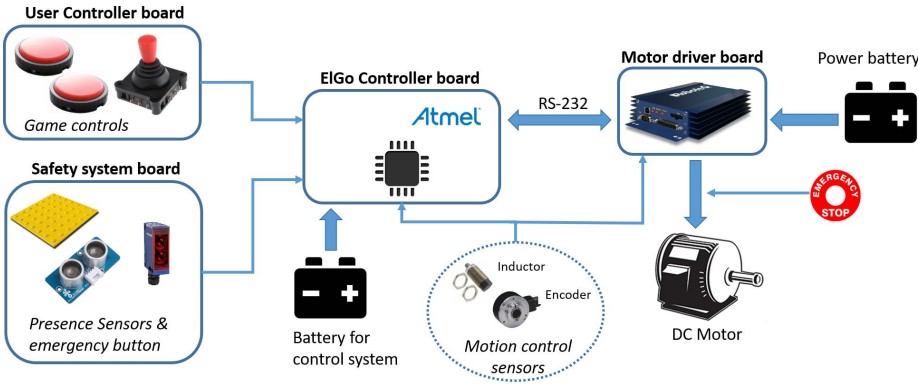

**Figure 4.** Functional block diagram of the electronic subsystem.

The first element is an off-the-shelf current driver produced by Roboteq Inc., Scottsdale, AZ, USA [35], which is used to power and control the DC motor; for later reference, it will be called the Motor Driver (MD). The MD incorporates a full-featured Proportional, Integral, Differential (PID) control algorithm. It adjusts the power output based on the difference measured between the desired position (set by the user's issued command) and the actual position (captured by the encoder). In particular, every 1 ms, the algorithm measures the actual motor position and subtracts it from the desired position to compute the position error. The resulting error value is then multiplied by a user-selectable Proportional gain ($P_{gain}$ = 2.0). The effect is to apply power to the motor that is proportional to the distance between the current and desired positions: when far apart, high power is applied, with the power being gradually reduced and stopped as the motor moves to the final position (maximum velocity 600 rpm). The Integral component performs a sum of the error over time. The value is then multiplied by the configured gain ($I_{gain}$ = 2.0) and added to the output. This helps to reach and maintain the exact desired position when the error would otherwise be too small to energize the motor using the proportional component alone. The Differential component of the algorithm computes the changes to the error from one ms time period to the next. It aims to give a boost of extra power when starting the motor due to changes to the desired position value. We selected $D_{gain}$ = 0 to avoid this power contribution. In addition to its main control function, the MD also has several useful features for monitoring and managing the performance of the controlled motor such as an instant current meter, a current limiter, and a passive or powered brake. Although it can be used with several interfaces, both analog and digital, we selected the RS 232 UART port for our purposes.

The second is the user interface board, featuring a joystick, four push buttons, and four mono 3.5 mm jack inputs suitable to connect almost every commercial switch designed for disabled users. This board can be connected both with a wired USB link or wireless, by means of a Class 1 Bluetooth point-to-point link established exploiting the embedded transceiver; for later reference, it will be called User Controller (UC). The developed UC is shown in Figure 5. Its sensor equipment allows it to support all three game modes provided in the ElGo system.

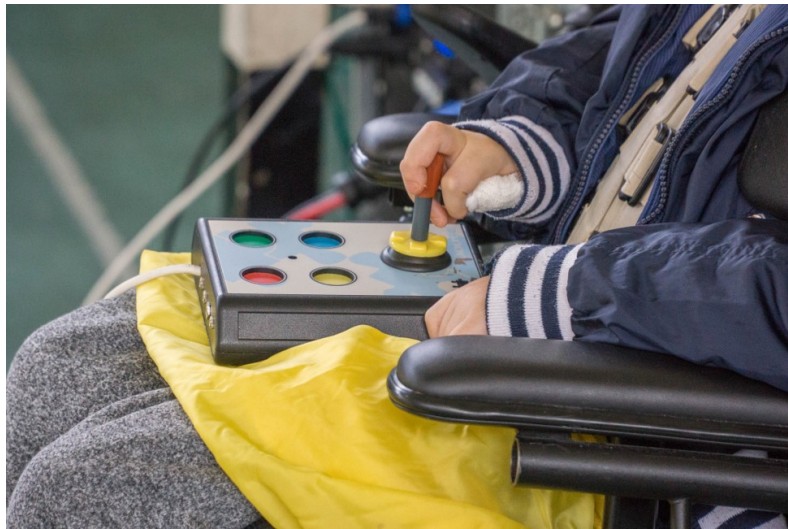

**Figure 5.** Universal controller enabling all three game modes.

The third board is the main controller, also referred to as the ElGo Controller (EC). Its task is to acquire all the signals sent by the user's input and sensors and combine them with the stored configuration (for instance the active game mode) in order to calculate the next motion-controlling action to be performed. Then, it sends the computed command to the MD to directly control the motor, and thus, the movement of the dummy. The EC also provides a simple user interface made of a rotary switch, four push buttons, and an

LCD display mainly dedicated to diagnostic and configuration purposes. For instance, it allows verifying the status of boards and sensors, changing the game mode on the fly, and calibrating the system at the startup.

The fourth and final board implements the safety system, which is a critical feature of the ElGo system. Because ElGo is designed for users to play together dynamically, it may happen during the game that a player gets too close to the goal line (i.e., too close to the dummy). Given the high mechanical torque of the motor and the direct-gear transmission, with no clutch inside, the displacement of the dummy could be very dangerous in case of impact with a person, in particular with a child. Thus, we developed a safety system based on data sensor fusion among several different types of sensors:

1.  Warning mat: it is basically a single-pole single-throw normally open OFF-Momentary-ON (SPST-NO), behaving similar to a push button. This mat can be activated by standing on it; a child or medium-sized animal can also be detected [36]. It is also used in industrial machinery to halt equipment when workers come too close to risky elements.
2.  Light grid: this is the same device used in lifts. It consists of an array of infrared beams with a high density and allows detection when an obstacle interrupts one or more light beams [37].
3.  Passive Infrared (PIR): these sensors are normally used in anti-intrusion systems. They are able to detect a variation of infrared beams within a large area [38], so they are useful to detect if a person, i.e., a player in our case, approaches the dangerous area.
4.  Ultra-Sonic (US): these sensors are composed of a transmitter and a receiver. The transmitter sends a US wave that is in the case reflected by an obstacle. Thus, by processing the reflected wave, it is possible to detect the presence of a specific object within a given range. In addition, by measuring the time-of-flight difference between send and return signals it is possible to detect the approximate distance of the obstacle. Given the simplicity of such calculations, these can be done by low-power microcontrollers [39]. For our purposes, US sensors are used to check if a person is standing inside the goal, beyond the goal line. This is useful because players need to go and take the ball inside the goal when they scored.

All the aforementioned sensors are powered and acquired by a dedicated board. Their outputs are then combined and sent to the EC. When at least one sensor is activated, the EC immediately halts the dummy. Movement will only restart when all sensors indicate that the conditions are safe to do so.

In addition, a big alarm button was introduced; this button is physically placed between the MD and the motor; when activated, any current flow in the motor is stopped immediately. It is not processed by either the EC or the MD, and thus, it is safe from any firmware bugs. After the alarm button is activated, the system must be reset in order to start operating again.

### 2.3.3. System Workload

The operation flow of the ElGo system in normal conditions (i.e., without pressing the big alarm button) is shown in Figure 6. The EC firmware executes a cycle that begins by querying the UC for the user's input commands. For safety purposes, if for any reason no command is received, a stop command is calculated. Otherwise, the user command is retrieved and considered for later calculations. Next, the EC verifies the safety conditions by probing sensors belonging to the safety system. If at least one of them reports a safety problem (i.e., is in active status), a stop command is calculated. Otherwise, the user command is elaborated with respect to the configured gaming mode, and the resulting final activation command is sent to the MD, which is responsible for managing ElGo's motor, and ultimately producing the motion of the dummy as requested by the user. In addition to this normal operating mode, the alarm button can be pressed at any time: since it is directly connected between the MD and the motor when it is activated, the MD is no longer able to supply power to the motor and the sequence of actions described above is interrupted.

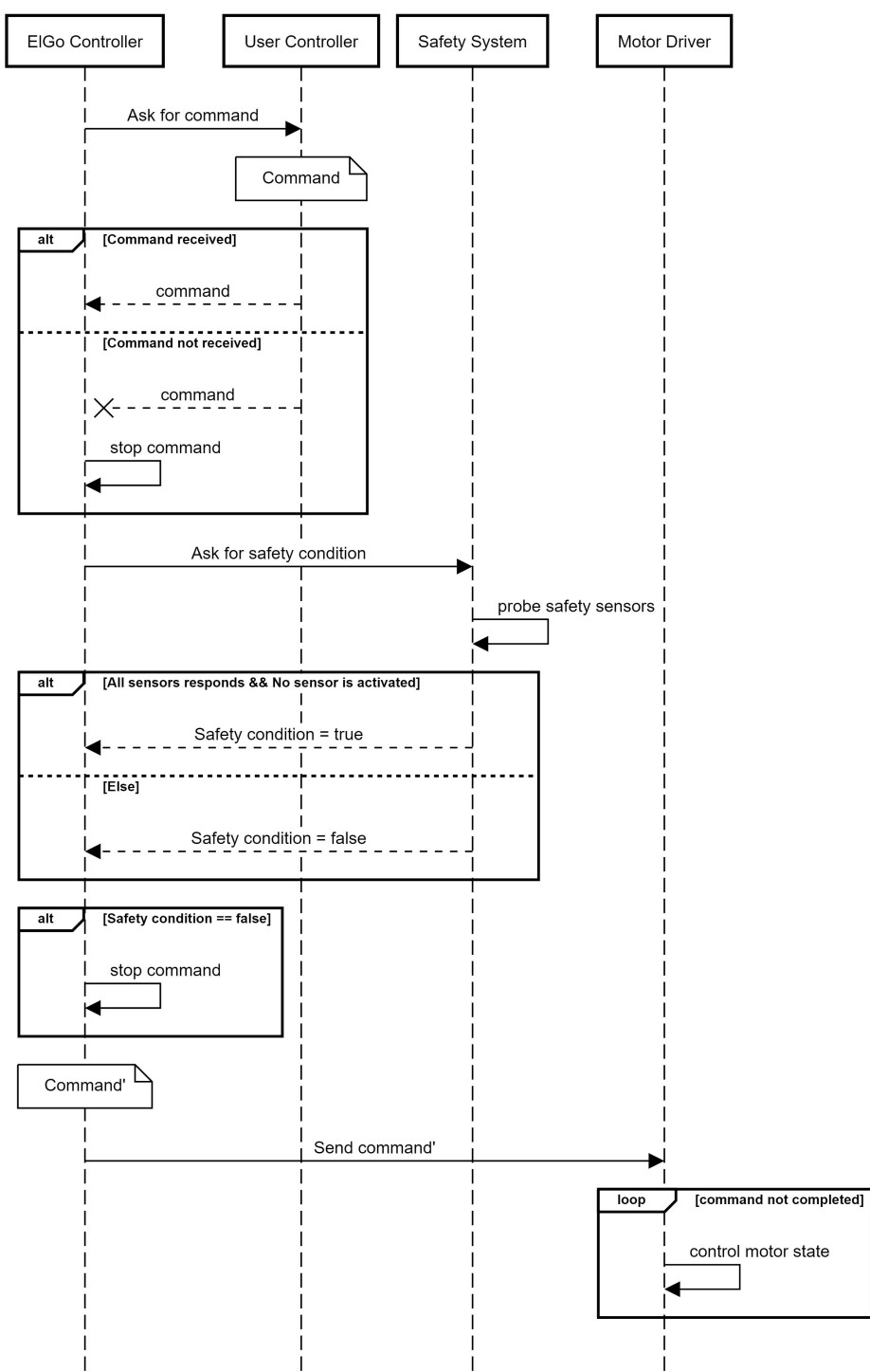

**Figure 6.** Sequence of actions executed for every system cycle.

### 2.4. Testing Procedure

The proposed ElGo system was initially tested with a restricted group of participants in a controlled environment to preliminarily evaluate whether the probability of saving a kick using ElGo is similar to that of a goalkeeper. To this end, we collected statistics regarding saves vs. total shots on goal.

Furthermore, a pilot testing was conducted with students and teachers of physical education from two high school classes to further investigate the outcome of the preliminary test, assess the approval by end users, and collect feedback from people without disabilities about the importance of including people with motor-skill disabilities in

team games. For this purpose, we used saving statistics, conducted non-structured interviews, and administered anonymous questionnaires to participants without disabilities. Appendixes A and B contain the questionnaires for students and teachers, respectively. Both questionnaires include ten questions, seven of which are multiple-choice and the remaining open-ended.

The pilot testing was extended by exploiting a public sporting event. The objective was to assess the system's robustness against extensive and prolonged usage and to collect additional insights from users.

All participants gave their informed consent for inclusion before they participated in the study. The specific setup of the tests and the involved participants are described in detail in the following section, along with the obtained results.

## 3. Results and Discussion

The overall performance of the system, its level of acceptance and usability, and the potential impact on the social integration of people with motor disabilities in sports activities were evaluated by field tests, interviews, and questionnaires. The following subsections present and discuss the obtained results.

### 3.1. Comparison with the Non-Professional Goalkeeper

In order to let a person with disabilities play at the same level as their non-disabled peers, ElGo was designed to be easy to use and competitive in a real scenario, providing similar performance compared to a human goalkeeper. To verify the achievement of these objectives, we conducted some comparison tests to assess if the probability of saving a kick using ElGo is similar to the human goalkeeper's one.

The test involved a total of six volunteer users aged between 16 and 20 years. We considered two disabled users affected by Duchenne syndrome and cerebral palsy, respectively, who played using ElGo, and four non-disabled users who alternately played both roles of physical goalkeeper and ElGo user. The latter also played the role of strikers, taking shots towards the goal when they were not engaged as goalkeepers. We tested two common situations in football: the first is a penalty, and the second is a dynamic action, i.e., when players are running and passing the ball to each other, waiting for the best occasion to kick the ball. In this case, we applied a restriction rule to prevent the attacker from getting too close to the goal, thus avoiding the safety system to activate and facilitate ElGo users to save goals. In particular, we imposed a radius of 6 m from the goal as the minimum shooting distance, which identifies a prohibited area that is easy to spot because it overlaps the penalty area of a five-a-side football pitch. In actual games, there is no need to impose this large prohibited area: the only constraint to ensure a fair game is to prevent players from stepping on the warning mat to prevent ElGo from stopping. Only kicks inside the net were analyzed. The result of these tests is reported in Table 2.

**Table 2.** Goal-saving statistics in case of penalties and during the dynamic game.

| User | Penalty with ElGo Save/Try | Penalty without ElGo Save/Try | Dynamic Action with ElGo Save/Try | Dynamic Action without ElGo Save/Try |
|---|---|---|---|---|
| Disabled user 1 | 11/30 | – | 15/30 | – |
| Disabled user 2 | 9/30 | – | 14/30 | – |
| Non-disabled user 1 | 18/30 | 12/30 | 21/30 | 21/30 |
| Non-disabled user 2 | 18/30 | 15/30 | 24/30 | 18/30 |
| Non-disabled user 3 | 15/30 | 16/30 | 22/30 | 18/30 |
| Non-disabled user 4 | 14/30 | 13/30 | 24/30 | 19/30 |

Despite the limited dataset, the test provides a preliminary result showing that the use of ElGo is effective in both game situations. In general, we observed better performance in dynamic action vs. penalty, as it is more difficult for the striker to score depending on the visible goalmouth and the position of ElGo. The system was easy to use for users with disabilities and allowed them to be competitive in the role of goalkeeper, in comparison

with a goalkeeper without disabilities. Indeed, ElGo reported a chance of saving a goal for a user with disabilities not so far from that of a player without disabilities engaged as the goalkeeper (33% vs. 46% on penalty and 48% vs. 63% in dynamic action, on average). This means that it can be truly competitive on the ground. Furthermore, we found that ElGo usually offers a higher probability of saving goals in the group without disabilities both in penalty and dynamic actions, as its reaction time is designed to be slightly faster than that of a non-professional goalkeeper and there is no fear of being hit by the ball, which sometimes plays an important role for non-professional goalkeepers.

### 3.2. Pilot Testing

A more extensive pilot testing was conducted in a real-life application context, aimed at confirming the performance achieved by the ElGo system during the preliminary test and collecting feedback from the participants.

The first scenario was at the gym of a high school. A group of 24 students without disabilities aged between 16 and 18, both male and female, was involved. In addition, the two boys with motor-skill disabilities already involved in the previous test were included, making a total of 26 participating players. Two teachers were also included in the sample. The setup of this test is shown in Figure 7a,b. The player with motor-skill impairments stands behind the goal sitting on their wheelchair so that the goal will also protect them from being hit by the ball. Then he controls the dummy moving left and right along the goal line, trying to save kicks performed by the other players. The test lasted approximately 3 h, with a pause of half an hour, and the users with disabilities played with ElGo both during a football match and a set of penalty kicks. In terms of goal-saving performance, the test reported results in line with the experimental data described above. In addition to the positive results on the technical side, we got some very positive ones on the user experience. Indeed, the tests confirmed the ElGo system's usability.

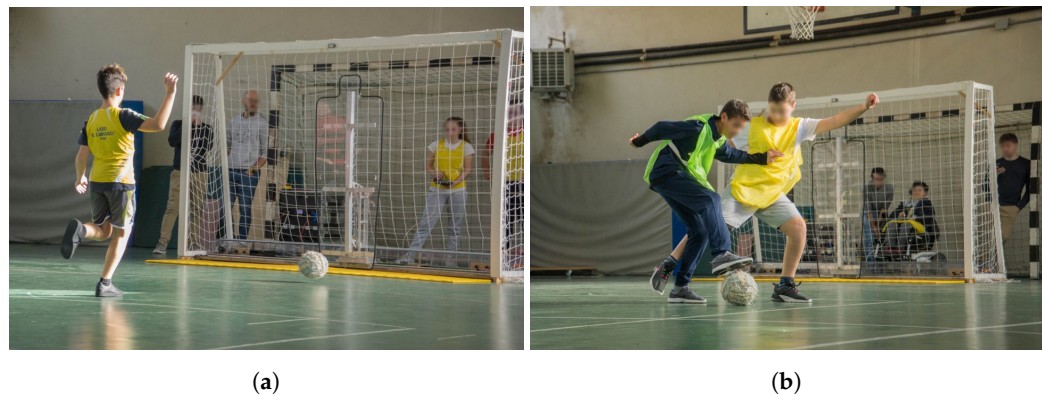

(**a**) (**b**)

**Figure 7.** Pictures of the first pilot testing session: (**a**) user and ElGo system and (**b**) construction of dynamic action.

Another pilot testing session was led during a sporting event in the central square of a town. The setup of this test is shown in Figure 8a,b. This test was open to anyone who wanted to try the system. On this occasion, we had roughly one hundred users, including people with different degrees of disabilities curious to try ElGo. The test lasted more than 4 h of continuous play, but considering the context, no performance statistics were collected.

During a total of around ten hours of testing, there were no faults in the electro-mechanical components nor in the electronic boards and sensors, demonstrating the high robustness of the system. Our tests showed also very good behavior of the safety system, facing only three faults, all due to the wrong positioning of the goal infrastructure.

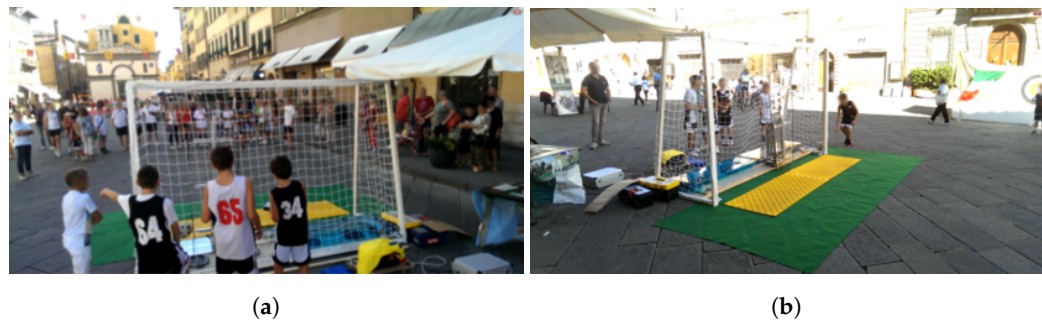

<table>
<tr><td>(**a**)</td><td>(**b**)</td></tr>
</table>

**Figure 8.** Pictures of the second pilot testing session: (**a**) user and friends with ElGo system and (**b**) ElGo system setup.

### 3.3. Approval by End Users

All the users with disabilities involved in the pilot testing were interviewed. Such interviews showed how well ElGo was accepted by them They really enjoyed ElGo, mostly because they never had the possibility to play at the same level with peers and because they saw a very impressive probability to save a goal, i.e., something that was really impossible before. They reported positive feedback about the usability of the system and the flexibility of user controllers. Moreover, they had very enthusiastic comments about ElGo and they would not stop to use it even after hours of playing. We believe that, even though ElGo is considered a piece of AT, it is actually perceived as a game, and thus a pleasant object to use, as opposed to a wheelchair or other aids.

All the users without disabilities were asked for feedback, using a questionnaire in the first pilot testing session and interviews in the second one. ElGo was readily also accepted by them since playing with/against it was really challenging and engaging. We also noticed another important thing during the open tests: kids without any impairment wanted to try ElGo as the goalkeeper, probably because they perceived it as a sort of videogame brought to reality. We believe this is very important for social inclusion, since it is unusual that people without disabilities would like to use pieces of AT; this provides a situation in which the person with disabilities can feel the same as their non-disabled peers.

### 3.4. Questionnaire for Users without Disabilities

During the first pilot testing session at the school, we dispensed two different questionnaires: one for the involved students and the other for teachers. The aim was to collect the impressions of users without disabilities on aspects regarding their perceptions of the ElGo system and its ability to include people with disabilities in sports and or recreational activities.

All students considered ElGo suitable for entertainment and 20 out of 24 responded positively to the question about the suitability of the system for team play. ElGo was mainly considered a mean to give every student the same opportunities to play and its use for facilitating the participation of students with disabilities was primarily considered entertaining. Almost all students (23) deemed the involvement of peers with disabilities by means of ElGo important. Table A1 reports in detail the collected responses to multiple-choice questions. In the open-ended questions, the participating students (7) expressed a general satisfaction and belief that playing together with disabled peers is very important. In addition, we received some critics about the dummy visibility: we resolved this problem by putting a colored rubber stripe on the dummy border.

Participating teachers agreed on the support provided by ElGo for team games and the possibility to use the system in activities of physical education at school. They pointed out that entertainment is the most important value to facilitate the participation of all students in educational activities. The usefulness of ElGo for socialization through sport and providing equal opportunities for all students were also reported. In open-ended questions, they underlined the importance of socialization and integration of people with

disabilities and one of them reported the issue related to the visibility of the dummy. The detailed results of closed-ended questions are shown in Table A2.

Ultimately, the collected responses suggest a general appreciation of the ElGo system and show how it can be an effective instrument for the social inclusion of people with motor-skill disabilities in sports and recreational activities in real-life situations, for example at school, in a public park, or football ground.

## 4. Conclusions

ElGo is a remotely controlled electronic goalkeeper designed to enable a user with motor-skill disabilities to play the role of the goalkeeper in a real, non-professional five-a-side football match. Thanks to the extensible human–machine interface, ElGo can be adapted to different needs of users with motor-skill impairments. To further improve usability, one possible solution is to extend the control by adding the possibility of using a Brain-Controlled-Interface helmet. By directly using the brain as an input source, this would increase the number of people able to interface with our system.

ElGo proved to be safe and have good usability from the user's perspective. From a technical point of view, ElGo showed a performance similar to that of a non-professional goalkeeper. Our pilot testing demonstrated the potential for use for people with motor-skill impairments, enabling them to play football with their peers, in real-life situations, and with competitive performance.

Almost all collected feedback were positive and showed how ElGo can be an effective instrument for the social inclusion of people with motor-skill disabilities. Furthermore, we noted a high level of interest in ElGo expressed by users without disabilities; their desire to engage with this disability aid can be beneficial to the self-esteem of individuals with disabilities. However, ElGo was specifically designed to allow people with disabilities to play football together with their peers, so the most relevant qualitative result we have achieved is the possibility to play football in a mixed way with a good level of engagement for everyone.

**Author Contributions:** Conceptualization, M.T., M.D. and L.F.; methodology, M.D., M.T. and L.B.; software, M.D., M.T., L.B. and F.P.; validation, L.F., L.B. and F.P.; formal analysis, M.D.; investigation, M.D., M.T. and F.P.; resources, M.T. and L.B.; data curation, M.D.; writing—original draft preparation, M.D., M.T. and F.P.; writing—review and editing, L.F.; visualization, M.D. and L.F.; supervision, L.F.; project administration, L.F.; funding acquisition, L.F. All authors have read and agreed to the published version of the manuscript.

**Funding:** This research was partially funded by LEO Club 108La, the company Dialog Semiconductors, and the Fondazione Banca del Monte di Lucca.

**Institutional Review Board Statement:** Not applicable.

**Informed Consent Statement:** Informed consent was obtained from all subjects involved in the study.

**Data Availability Statement:** The data presented in this study are not publicly available due to privacy restrictions.

**Acknowledgments:** The authors would like to thank every person that collaborates at the ElGo project: R. Roncella, G. Fantechi, F. Iacopetti, L. Sciurti, M. Lombardi, C. Mattaliano, F. Bernardo, P. Neri, A. Benini, and A. Frello from University of Pisa and the Teachers and Students of Liceo Coluccio Salutati of Montecatini Terme and of Liceo Carducci of Pisa.

**Conflicts of Interest:** The authors declare no conflict of interest.

## Abbreviations

The following abbreviations are used in this manuscript:

| | |
|---|---|
| ADC | Analog-to-Digital Converter |
| AT | Assistive Technologies |
| CRPD | Convention on the Rights of People with Disabilities |

| DC | Direct current |
| EC | ElGo Controller |
| ElGo | Electronic Goalkeeper |
| LCD | Liquid Cristal Display |
| MD | Motor Driver |
| PID | Proportional, Integral, Differential |
| PIR | Passive Infra Red |
| RTOS | Real-Time Operating Systems |
| SPST-NO | Single-Pole Single-Throw Normally-Open |
| UC | User Controller |
| US | Ultra Sound |
| RL | Reinforcement Learning |

**Appendix A**

The questionnaire for students without disabilities.

1.　*Do you think that the electronic goalkeeper is suitable for entertaining friends and mates?*

　☐　Yes
　☐　No

2.　*Do you think that the electronic goalkeeper is suitable for the team game?*

　(a) Strong agree

　(b) Agree

　(c) Neither agree nor disagree

　(d) Disagree

3.　*How do you consider the electronic goalkeeper inclusion in the school to facilitate the participation of students with disability?*

　(a) Essential

　(b) Entertaining

　(c) Useless

4.　*How important do you think is the involvement of students with disabilities by means of the electronic goalkeeper?*

　(a) A great deal

　(b) Very important

　(c) Ordinary

　(d) Needless

5.　*The inclusion of the electronic goalkeeper at school can be useful for which purpose? (multiple answers allowed)*

　(a) Nothing

　(b) To have fun and to give every student the same opportunities to play

　(c) To encourage socialization through sport

　(d) To include all students in sports

6.　*Do you think the electronic goalkeeper can be used during the activities of Physical Education?*

　☐　Yes
　☐　No

7.  *Do you think that the electronic goalkeeper should be used for recreational purposes only?*

    ☐ Yes
    ☐ No

8.  *Give briefly some criticism of the project (if you found)*

9.  *Give briefly what are the strengths and benefits of the project (if you found)*

10. *Suggestions/Comments/Requests*

**Table A1.** Results of the students' questionnaire.

| Question | Answer Options | Results |
|----------|---------------|---------|
| 1 | Yes/No | 24/0 |
| 2 | a/b/c/d | 1/19/4/0 |
| 3 | a/b/c | 8/16/0 |
| 4 | a/b/c/d | 22/1/1/0 |
| 5 | a/b/c/d | 0/16/0/ 8 |
| 6 | Yes/No | 24/0 |
| 7 | Yes/No | 6/18 |

## Appendix B

The questionnaire for teachers.

1.  *Is the electronic goalkeeper intended only as a support for recreational activities?*

    ☐ Yes
    ☐ No

2.  *Do you think that the electronic goalkeeper is to be considered a valuable additional support for the team game that can be used during the teaching of Physical Education?*

    (a) Strong agree

    (b) Agree

    (c) Neither agree nor disagree

    (d) Disagree

3.  *What educational value can be attributed to the electronic goalkeeper as a strategic tool to facilitate the participation of all students in educational activities?*

    (a) Essential

    (b) Entertaining

    (c) Useless

4.  *Do you think that the inclusion of students with disabilities by means of the electronic goalkeeper is important?*

    (a) Strong agree

    (b) Agree

    (c) Neither agree nor disagree

    (d) Disagree

5. *For what purposes do you think that the electronic goalkeeper can be useful at school? (multiple answers allowed)*

   (a) Nothing

   (b) To offer all students the same opportunities to express themselves

   (c) To encourage socialization through sport

   (d) To include all students in sports

6. *Do you think that the electronic goalkeeper can be used during activities of physical education in schools?*

   ☐ Yes
   ☐ No

7. *Do you think that the electronic goalkeeper can be an essential part of multi-disciplinary teaching strategies?*

   ☐ Yes
   ☐ No

8. *Express briefly some criticism or weaknesses of the project (if you found)*

9. *Express briefly what are the benefits of the project (if you found)*

10. *Suggestions/Comments/Other educational projects that see the use of the instrument showing aims and objectives*

**Table A2.** Results of the teachers' questionnaire.

| Question | Answer Options | Results |
|---|---|---|
| 1 | Yes/No | 0/2 |
| 2 | a/b/c/d | 0/2/0/0 |
| 3 | a/b/c | 0/2/0 |
| 4 | a/b/c/d | 0/1/1/0 |
| 5 | a/b/c/d | 0/1/1/0 |
| 6 | Yes/No | 2/0 |
| 7 | Yes/No | N/A |

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
