# Peer review of "Remotely Controlled Electronic Goalkeeper: An Example of Improving Social Integration of Persons with and without Disabilities"

_applsci, doi:10.3390/app13116813_

Round 1

Reviewer 1 Report

This manuscript addresses a current issue and poses an interesting research question. I do however raise some concerns in both the method, analysis and the reporting of the results that I think require some attention, as well as several substantial gaps in the writing that compromise the contribution this manuscript could make to the field.

Specific recommendations for improvements are provided below.

Abstract

·        The writing is informal in places, more as you’d talk than you would write. For example, “there is a sort of barrier” – there either is, or there isn’t a barrier.

·        Struggling to follow the third sentence; “not only during dedicated events” (what are these events?); “…but also during everyday life” – but you opened with the fact participation is low, so is it really in everyday life? What are you referring to here, more formal sporting groups?

·        Following you state “…while the mixing of people with and without disabilities in sports activities is a key factor of social inclusion”. This seems like a new sentence, or it seems unfinished, just doesn’t seem to flow from what went before.

·        “His/her” – this in itself is exclusionary to people that identify outside of the binary of male and female genders and should be changed.

·        Are they “mates” – again that is more informal language; wonder if “peers” would be more appropriate.

·        I think you can just state “that lets a person play the role of goalkeeper” – as you have already mentioned the Assistive Technology is for people with motor-skill impairments.

·        Is the word ‘live’ necessarily? What would a non-live non-professional football match look like?

·        “with the same performances as a human goalkeeper” – but you mentioned the AT allows a person with motor-skill impairments to paly the role of the goalkeeper; so human seems like the wrong word here; do you mean a goalkeeper without motor-skill impairments?

·        However, when you describe the AT, it doesn’t seem like the person is playing the role of the goalkeeper; rather it’s the AT acting as the goalkeeper, so in which case the earlier part about ‘playing the role of’ doesn’t seem accurate.

·        Rather than “testing campaign” perhaps better to say “results of pilot testing”?

·        You switch in the last couple of sentences to “players with severe mobility impairments” (is that the same as motor-skill impairments); and then “people with disabilities” – but you only tested with motor-skill impairments so not sure it is accurate to generalise.

Introduction

·        There is plenty of research to suggest that social inclusion is worse for people with disabilities; so not sure on the accuracy of the second sentence. What do you mean by “…and now has become a fact in many human real situations”? As opposed to virtual reality? Not sure “fact” is the right word. Certainly the CRPD outlines the importance of it…but its far from a fact, there is a long way to go, hence your research.

·        Following your next sentence needs changing too; social functioning, activities of daily living and employment opportunities can be impacted. There is a plethora of research which shows people with disabilities have lower employment rates, lower levels of social participation. Also, for people with more profound disabilities not sure it would be accurate to say they live independently, as they need ongoing support to facilitate participation. And as you go on to say, more work needs to be done. But what do you mean by “to reach a really full social inclusion” It contradicts all you have said before, but not sure what “really full” looks like.

·        I don’t understand the opening sentence on line 26. Its also vague “specific actions” – but only cite architectural barriers. The tireless work of who, and in what countries are people reaching a higher social status? What does that even mean?

·        You go from the UNCRPD to technological solutions, it lacks flow, quite the jump. The link would be to state that article four includes the development of, and availability and use of new technologies, including assistive technologies….

·        Please rephrase “suffer from different types of disabilities” – maybe “live with” would be more appropriate.

·        “As a matter of fact”; “To this aim” – is informal writing, please rephrase.

·        You shift between people with disabilities and “disabled people” – keep consistent.

·        Not sure its accurate to state that AT means people with disabilities “can do most of the things that non-disabled people do”; rather AT is designed to improve the functional capabilities of people with disabilities; can help people perform tasks that might otherwise be difficult or impossible.

·        “We noticed a general lack of social integration” – how did you notice this? You are citing someone else’s research here, so do you mean that research has demonstrated this.

·        Why is socialisation for younger people “more important than for any adult”

·         “girls/boys” - it would be great if the authors could be respectful of people with and without disabilities that identify outside of the binary, yet it is often not explored, or even recognised, so I think as researchers we have a role to play to remove the inequities.

·        “Practicing sports” – or do you mean “Participation in sports”?

·        “will improve well-being” – is “can improve well-being” more accurate?

·        “Sometimes this lack is due to the poor awareness” – a word is missing somewhere here, maybe ‘lack of participation’

·        Please rephrase the sentence that starts on line 45. Firstly it is written informally, again, “his/her”, and “mates”; and I gasped when I read it, I just don’t think it is at all respectful to people with disabilities, very badly phrased.

·        The sentence that starts on line 49 I can’t follow, “mainly concerns on dedicated activities” – “sporting assemblages” – what does it all mean?

·        “This paper instead” – informal writing again; and another example of shifting language (“disabled person”). You have previously used “mates”, here “friends” – I’d again suggest ‘peers’ is more appropriate. Some of this also seems like methods rather than an introduction. Here it is about introducing the why, and your aims.

·        I think it would also be helpful to talk about people motor-skill impairments, for example, that this includes people with cerebral palsy, multiple sclerosis etc; and the support needs – why it is difficult for them to participate in sport, the barriers that have been identified etc. Currently this doesn’t come in until around line 154.

Materials and methods

·        Please check all language used when talking about people with disabilities, as “even severe ones”; “he users’ peculiar characteristics” – there are better words you could use that would be more respectful.

·        I would suggest referring to the figure of the EIGo earlier; and the image in Figure 6 is helpful to understand how it all works. From the writing I am not understanding the concept and I think this could help to better explain the role of the person using the AT.

·        I think what you are describing here, is a co-design and development phase; followed by pilot testing, and your paper restructured to reflect the two components. For example, some of the information under ‘system requirements’ is analysis / results of the co-design work, and should come in later, or at least after you describe what the EIGo is.

·        “As said before” – informal, needs rephrasing.

·        Why did you decide to just test with boys? Some justification might be helpful here.

·        How did you group the boys into skill level? Are the weights reported averages, or did all boys rated as skilled weigh 62kg for example?

·        It seems after you did the testing with boys, you have instead used the data obtained from professional footballers, as I can’t see how else you took the value of 20 m/s, as the highest in Table 1 is “11,5”. I am not sure what the initial testing was used for.

·         

Results and Discussion

·        Why were there only two people with disability (and why only Duchenne syndrome and cerebral palsy chosen); but 6 people without disability? The recruitment of participants should be mentioned in the methods – was there consent, was ethics obtained, again all methods but not mentioned.

·        Were they matched in terms of age, weight to people without disabilities, or does that not matter?

·        You state you were testing in real situations, but “applied a restriction rule to prevent the attacker from getting too close to the goal” – that wouldn’t be able to happen in a ‘real situation’ – not aware of a rule in the game about that, so it sounds like some modifications were required for safety, and how would that work in an actual game?

·        Not sure how testing 2 vs 6 is adequate to show they can play at the same level; and what comparisons did you do to demonstrate there was no “negative trend”.

·        Testing campaign – does seem more like pilot testing. But again the majority of the sample you tested with did not have a disability, only 2 of the 24 did. Given the aims of the study were to allow a person with motor-skill impairment to play, not sure how such a small sample can demonstrate this. It more demonstrates how the system can be used, and the potential for use for people with motor-skill impairments?

·        Further, the second test you mention on line 373 “we had many non-disabled people” – what did this sample look like, how many people were there with and without disabilities? No performance statistics were collected, and if similarly no descriptives were collected, I’m not sure I understand what this test shows, other than ‘levels of interest/curiosity’?

·        When the system was tested for a further 10 hours, who did this testing, under what conditions, and by whom?

·        You mention “all the disabled users were interviewed after the game”. What game is this referring to, the sporting event? How many interviews, and how were the interviews analysed. Usually interview data requires thematic analysis, but this isn’t detailed.

·        You mention “EIGo was readily also accepted by the other non-disabled players” – so were they also interviewed, this wasn’t previously mentioned.

·        The questionnaire you mention on line 400, is partly methods, rather than results. I wonder whether the questionnaire should be briefly described here, and the questionnaire itself be included as an appendix (supplementary file), rather than in the body of the manuscript?

·        There were 26 people, but only 24 respondents is that correct?

·        The presentation of the results in Table 3 could be improved, for example, question number becomes; Suitability for entertaining friends and peers; and then you present the number that responded Yes (the alternative is just what is left over, in this case 0); suitability for team game – you just present the total that selected Agree or Strongly agree; etc etc.

·        You introduce the questionnaire for teachers, but this wasn’t mentioned in the methods, how many teachers did you recruit? Similar comments regarding the reporting of this questionnaire to the above comments.

·        This section mostly seems results, I am struggling to see where the discussion is; that’s the “so what does this all mean”; it’s what can be learnt from your study

Conclusion

·        Line 502 – missing skill “motor disabilities”?

·        You state EIGo can be used by “users with various types of disabilities” – but not sure you demonstrated this given the sample you used to develop and pilot it.

·        Not sure based on the data collected and how you have been able to conclude “we believe that this desire from non-disabled users to have something that the disabled users have can be beneficial for the self-esteem of disabled people” – is it simply a belief, or something you can state based on data collected?

The writing is often quite informal, written as you would speak rather than write for a journal.  The use of language is also quite exclusionary in terms of gender identify, and at times disrespectful to people with disabilities. Further detail is provided in the comments above.

Reviewer 2 Report

Strong aspects:

The authors present a system for goal keeping when playing games that can be controlled by disabled persons with their hands. As results show this is an interesting tool to use. Detailed presentation of the system is given.

 Weak aspects:

The scientific benefit of the presented work is limited. The impact of this work would be stronger if the artificial goalkeeper would be controlled directly by the brain using a BCI helmet.

 Comments to the authors:

 - The detailed questionnaire is not useful. Some ideas related to this can be presented in a few lines.

- The pictures in Fig. 6 and 7 should have higher resolution. The EIGo system is not clearly visible.

- In order to simplify the visualization of the results the for the non-disabled users with and without EIGo should be presented on the same row. I suggest splitting in 2 the columns 'Penalty' and 'Dynamic action'. 

- 'quite skilled' in the Table 1 should be replaced by 'medium skilled'

- 'a little faster' in line 164 should be replaced by 'faster'

- In line 178 in the expression '13m/s2' the '2' should be superscript.

-

Reviewer 3 Report

This paper presented the Electronic Goalkeeper (ElGo) to play the role of the goalkeeper in a five-a-side live football match. However, some following problems need to be discussed:

1. The method of handling Delay (Section 2.2.2) needs to be pointed out and compare with the previous technique, such as Time Varying Delay in Bilateral Teleoperators https://ietresearch.onlinelibrary.wiley.com/doi/full/10.1049/cth2.12155;

2. The general flowchart of this system should be given;

3. The control method should be presented in section 2.3.2;

4. The user experience (Section 2.2.4) is mentioned in this work. Therefore, some related methods need to be discussed and compared, such as Neural Network and Learning method in https://ieeexplore.ieee.org/abstract/document/9967790, https://www.sciencedirect.com/science/article/abs/pii/S0019057822001495

The English quality is appropriate for Publication in Journal
